# Extended ensemble simulations of a SARS-CoV-2 nsp1–5'-UTR complex

**Shun Sakuraba**  [1] *, **Qilin Xie** [2], **Kota Kasahara** [3], **Junichi Iwakiri** [4], **Hidetoshi Kono** [1]

**1** Institute for Quantum Life Science, National Institutes for Quantum Science and Technology, Kizugawa, Japan, **2** Graduate School of Life Sciences, Ritsumeikan University, Kusatsu, Japan, **3** College of Life Sciences, Ritsumeikan University, Kusatsu, Japan, **4** Graduate School of Frontier Sciences, The University of Tokyo, Kashiwa, Japan

* sakuraba.shun@qst.go.jp

**Data Availability Statement:** The data and code to reproduce the research is included in the Supporting information and BSMA archive (https://bsma.pdbj.org/entry/26).

## Abstract

Nonstructural protein 1 (nsp1) of severe acute respiratory syndrome coronavirus 2 (SARS-CoV-2) is a 180-residue protein that blocks translation of host mRNAs in SARS-CoV-2-infected cells. Although it is known that SARS-CoV-2's own RNA evades nsp1's host translation shutoff, the molecular mechanism underlying the evasion was poorly understood. We performed an extended ensemble molecular dynamics simulation to investigate the mechanism of the viral RNA evasion. Simulation results suggested that the stem loop structure of the SARS-CoV-2 RNA 5'-untranslated region (SL1) binds to both nsp1's N-terminal globular region and intrinsically disordered region. The consistency of the results was assessed by modeling nsp1-40$S$ ribosome structure based on reported nsp1 experiments, including the X-ray crystallographic structure analysis, the cryo-EM electron density map, and cross-linking experiments. The SL1 binding region predicted from the simulation was open to the solvent, yet the ribosome could interact with SL1. Cluster analysis of the binding mode and detailed analysis of the binding poses suggest residues Arg124, Lys47, Arg43, and Asn126 may be involved in the SL1 recognition mechanism, consistent with the existing mutational analysis.

## Author summary

The pandemic of COVID-19 is still rampant all over the world as of 2021 June. SARS-CoV-2 (severe acute respiratory syndrome coronavirus 2), the causative pathogen of COVID-19, encodes a protein called nsp1 (nonstructural protein 1), which modulates and hijacks the ribosome of the infected host cells. With nsp1, infected human cells selectively translate SARS-CoV-2's RNA, which increases the virus reproduction efficiency while evading the host immunity. Though it has been known that nsp1 recognizes characteristic stem-loop structure at 5'-end of SARS-CoV-2's RNA (called SL1), the molecular mechanism underlying the recognition has been poorly understood. We investigated the mechanism of selective translation using the all-atom molecular dynamics simulation of nsp1-SL1 complex. Our simulation results suggest that the binding between nsp1 and SL1 is multi-modal. The results also imply that both the N-terminal globular part and the

**Funding:** SS was supported by a Grant-in-Aid for Early-Career Scientists from the Japan Society for the Promotion of Science (JSPS; https://www.jsps. go.jp/english/), Japan (JP16K17778), by Grants-in-Aid for Scientific Research (A) from the JSPS (JP16H02484 and JP21H04912), and by a Grant-in-Aid for Scientific Research on Innovative Areas from the Ministry of Education, Culture, Sports, Science and Technology (MEXT; https://www. mext.go.jp/en/; JP19H05410). KK was supported by a Grant-in-Aid for Scientific Research (C) from the JSPS (JP20K12069). JI was supported by a Grant-in-Aid for Scientific Research (C) from the JSPS (JP20K12041). HK was supported by by Platform Project for Supporting Drug Discovery and Life Science Research (Basis for Supporting Innovative Drug Discovery and Life Science Research (BINDS)) from AMED under Grant Number JP21am0101106, Agency for Medical Research and Development (AMED; https://www. amed.go.jp/en/), Japan. The funders had no role in study design, data collection and analysis, decision to publish, or preparation of the manuscript.

**Competing interests:** The authors have declared that no competing interests exist.

C-terminal flexible tail of nsp1 are involved in the binding. The residues involved in nsp1-SL1 binding coincides with the known mutant analyses of SARS-CoV-1 and SARS-CoV-2, as well as experimental evidence about nsp1-ribosome interactions.

## Introduction

SARS-CoV-2 (severe acute respiratory syndrome coronavirus 2) belongs to *Betacoronaviridae*, and is the causative pathogen of COVID-19. Nonstructural protein 1 (nsp1) resides at the beginning of SARS-CoV-2's genome, and it is the first protein translated upon SARS-CoV-2 infection. After self-cleavage of open reading frame 1a (orf1a) by an orf1a-encoded protease (nsp3; PLpro), nsp1 is released as a 180-residue protein. SARS-CoV-2 nsp1 is homologous to nsp1 of SARS-CoV-1, the causative pathogen of SARS, sharing 84% sequence identity with the SARS-CoV-1 protein. Nsp1 functions to suppress host gene expression [1–6] and induce host mRNA cleavage, [1, 2, 7–9] effectively blocking translation of host mRNAs. The translation shutoff hinders the host cell's innate immune response including interferon-dependent signaling. [1, 10] Multiple groups have recently reported cryogenic electron microscopy (cryo-EM) structures of SARS-CoV-2 nsp1–40*S* ribosome complexes. [11–13] The structural analysis showed that two $\alpha$-helices are formed in the C-terminal region (153–160, 166–179) of nsp1 and binds to the 40*S* ribosome. These helices block host translation by shutting the ribosomal tunnel used by the mRNA. This blockade inhibits the formation of the 48*S* ribosome pre-initiation complex, which is essential for translation initiation. [3, 13] But while nsp1 shuts down host mRNA translation, it is known that the viral RNAs are translated even in the presence of the nsp1, and that they evade degradation. [2–4]

These mechanisms force infected cells to produce only viral proteins instead of normal host cell proteins; indeed, in a transcriptome analysis, 65% of total RNA reads from Vero cells infected with SARS-CoV-2 were mapped to the viral genome. [14] It has also been shown that nsp1 recognizes the 5'-untranslated region (5'-UTR) of the viral RNA [4, 6, 12] and selectively enables translation of RNAs that have a specific sequence. The first stem loop in the 5'-UTR [4, 6, 15] has been shown to be necessary for translation initiation in the presence of nsp1. Specifically, with SARS-CoV-1, [4] bases 1–36 of the 5'-UTR enable translation of viral RNA; with SARS-CoV-2, bases 1–33 [15] or 1–40 [6] of the 5'-UTR of SARS-CoV-2 enable translation. However, the precise molecular mechanism remains poorly understood.

In the present research, therefore, our aim was to accumulate information about the molecular mechanism by which SARS-CoV-2 RNA evades nsp1. As a first step, we focused on how and where SARS-CoV-2 5'-UTR binds to nsp1, and tackled the problem from computational simulations. We modeled and simulated a complex comprised of SARS-CoV-2 nsp1 and the SARS-CoV-2 5'-UTR's first stem loop using extended ensemble molecular simulations. The simulations suggested the importance of the nsp1's C-terminal disordered region as well as that of the globular region. The binding preference of the 5'-UTR onto nsp1 was assessed, and its consistency to the current ribosome-nsp1 model was investigated to further confirm the simulation results.

## Materials and methods

### Overview

We constructed a complex of nsp1 and 5'-UTR of SARS-CoV-2 RNA and performed simulations to investigate the mechanism behind the self-evasion of the nsp1's translation shutoff.

Nsp1 has an intrinsically disordered region (IDR) and is considered to bind to the RNA. However, it has generally been considered that RNA-protein complexes are difficult to simulate because structures tend to be trapped around the initial configurations in a reasonable simulation time, due strong charge-charge interactions between the RNA and the positively charged protein residues. To ease the problem, we performed an extended ensemble simulation. In extended ensemble simulations, modified energy functions are used to sample various possible structures of complexes. The effect of modified energy functions can be statistically removed in the post-process phase (with a procedure called reweighting), thereby enabling us to obtain structures of the nsp1-RNA complex at the given temperature in a comparably shorter simulation time than the conventional molecular dynamics (MD) simulations. After performing the simulation, we analyzed the trajectory to investigate which residues in nsp1 are contacting RNA and how the structure is formed.

## Simulation setup

Nsp1 is a partially disordered 180-residue protein, in which the structures of residues 12–127 and 14–125 have been solved by X-ray crystallography in SARS-CoV-1 and SARS-CoV-2, respectively. The structures of other residues (1–11, 128–180) are unknown, and residues 130–180 are thought to be an IDR. [16, 17] We constructed the SARS-CoV-2 nsp1 structure using homology modeling based on the SARS-CoV-1 nsp1 conformation (Protein Data Bank (PDB) ID: 2HSX [16]). Modeling was performed using MODELLER. [18] We noted that SARS-CoV-1 nsp1 and SARS-CoV-2 nsp1 are aligned without gaps. The structure of the IDR was constructed so as to form an extended structure. For nsp1, we used the AMBER ff14SB force field [19–22] in the subsequent simulations.

The initial structure of the RNA stem was constructed using RNAcomposer. [23, 24] Bases numbered 1–35 from the SARS-CoV-2 reference genome (NCBI reference sequence ID NC_045512.2) [25] were used in the present research. This sequence corresponds to the first stem loop of the SARS-CoV-2 RNA 5'-UTR. Hereafter, we will call this RNA "SL1." SL1 was capped by 7-methyl guanosine triphosphate (m7G-ppp-). The first base (A1) after the cap was methylated at the 2'-O position to reflect the viral capped RNA. Charges and bonded force field parameters for these modified bases were respectively prepared using the restrained electrostatic potential (RESP) method [26] and analogy to existing parameters. For SL1, we used a combination of AMBER99 + bsc0 + $\chi$OL3. [19, 20, 27, 28] To maintain the structural stability of the stem loop, we employed distance restraints between the G-C bases. Specifically, between residues G7–C33, G8–C32, C15–G24 and C16–G23, distance restraints were applied such that the distances between the N1, O6 and N2 atoms of guanosine and the N3, N4 and O2 atoms of cytidine, did not exceed 4.0 Å. Between these atoms, flat-bottom potentials were applied, where each potential was zero when the distance between two atoms was less than 4.0 Å, and a harmonic restraint with a spring constant of 1 kJ mol$^{-1}$ Å$^{-2}$ was applied when it exceeds 4.0 Å. We used `acpype` [29] to convert the AMBER force field files generated by AmberTools [30] into GROMACS. Parameter files are presented in S1 File.

The nsp1 and SL1 models were then merged and, using TIP3P [31] water model with Joung-Cheatham monovalent ion parameters [32] (73,468 water molecules, 253 K$^+$ ions, 209 Cl$^-$ ions), were solvated in 150 mM KCl solution. The initial structure is presented in Fig 1A. A periodic boundary condition using a rhombic dodecahedron unit cell was used with a size of *ca.* 140 Å along the X-axis. Note that we started the simulation from the unbound state; that is, nsp1 and SL1 were not directly in contact with each other. The total number of atoms in the system was 224,798. After preparing the system we also prepared the system without SL1 by removing it from the nsp1-SL1 initial structure (the total number of atoms was 223,598).

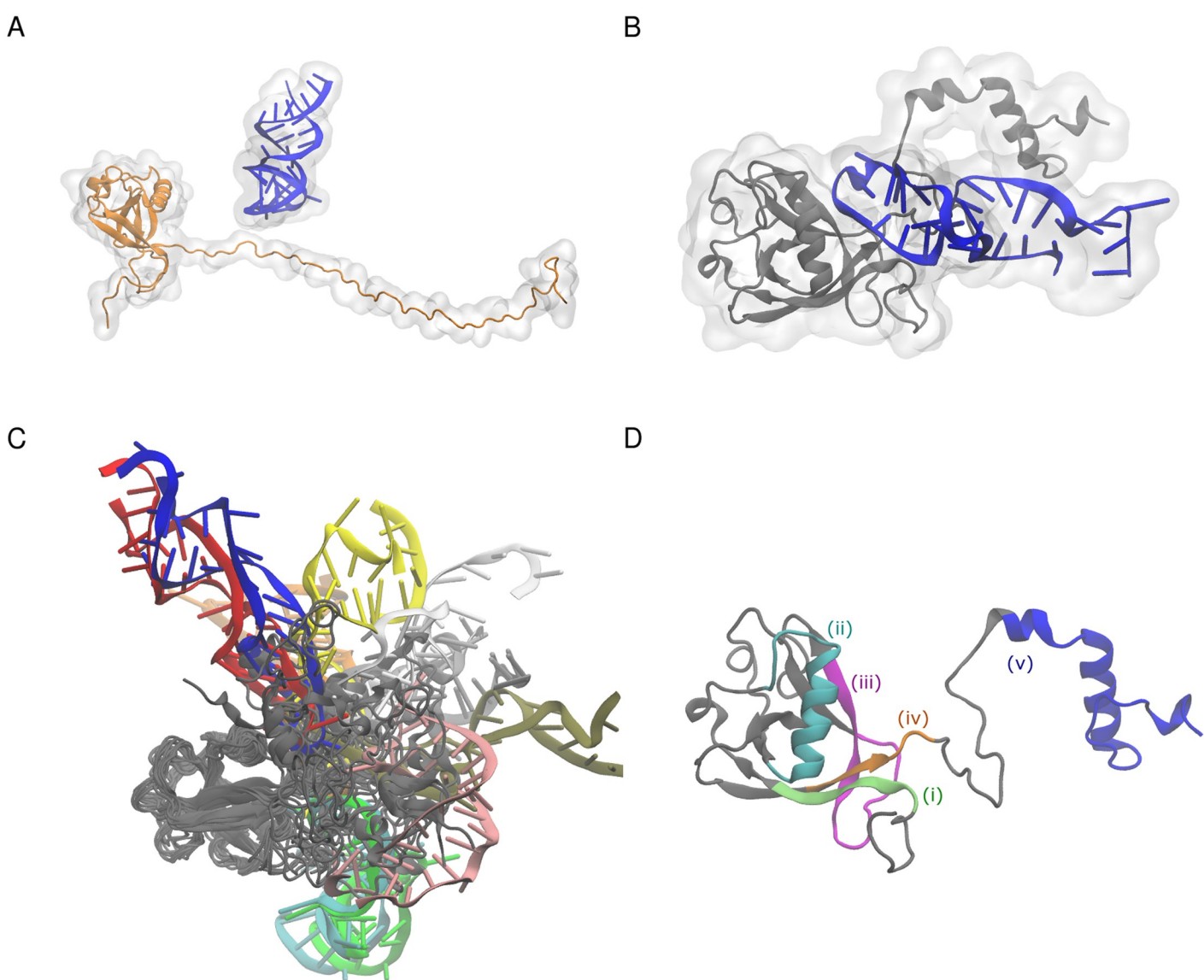

**Fig 1. Structures of nsp1.** (A) Initial structure before starting the simulation. (B) Structure of the complex at 50 ns in the 0th replica (i.e., the simulation with the unscaled potential). (C) Structures from superimposition of 20 representative snapshots of the nsp1-SL1 complex. Snapshots were obtained from a weighted random sampling. Different snapshots from SL1 are colored differently. (D) Nsp1 segmentation used in the analysis: (i) residues 1 to 18, green; (ii) residues 31 to 50, cyan; (iii) residues 74 to 90, magenta; (iv) residues 121 to 146, orange; (v) residues 147 to 180, blue.

Although it is possible to perform a MD simulation of an nsp1-SL1 complex, due to the excessive charges on both molecules, the model tends to be trapped around the initial configuration of the complex in conventional MD simulations. Authors have previously shown that the sampling for nucleic acid–protein systems can be effectively solved by extended ensemble simulations. [33–35] In this work, we used replica exchange with solute tempering (REST) version 2 to sample various configurations of SL1 and the nsp1 IDR. [36] In REST2, the simulations are performed so that specific residues (called a "hot" region) have weaker interactions with others than the conventional MD simulations. This modification to the potential function prevents the simulation to be trapped around the initial configuration. We set both the disordered region (nsp1 1–11 and 128–180) and the entire SL1 as the "hot" region of the REST2

simulation. Therefore, even though some base pairs of SL1 were restrained, the interactions between nsp1 and SL1 as well as SL1 and solvent were scaled in the REST2 simulation, allowing broader configurations to be sampled. Note that in addition to the charge scaling for nsp1 and SL1, we also scaled the charges of counter-ions to prevent unneutralized system charge in the Ewald summation. The total number of replicas used in the simulation was 192. The replica numbered 0 corresponds to the simulation with the unscaled potential. In the final replica (numbered 191), nonbonded potentials between "hot"-"hot" groups were scaled by 0.25. Exchange ratios were 53–78% across all replicas. To prevent numerical errors originating from the loss of significant digits, we used a double-precision version of GROMACS as the simulation software. [37] We also modified GROMACS to enable the replica exchange simulation with an arbitrary Hamiltonian. [38] The patch representing modifications is supplied in S2 File.

The simulation was performed for 50 ns (thus, 50 ns×192 = 9.6 μs in total), and the first 25 ns were discarded as the equilibration time. The simulation was performed with NVT and the temperature was set to 300 K. The temperature was controlled using the velocity rescaling method. [39] The timestep was set to 2 fs, and hydrogens attached to heavy atoms were constrained with LINCS. [40] Similar to the nsp1-SL1 complex, we also performed the simulation of nsp1 only (wihtout SL1) with exactly the same condition for 50 ns (another 9.6 μs simulation in total). Simulation input files and trajectories for the 8 lowest numbered replicas are deposited at Biological Structure Model Archive (BSMA; entry ID 26) https://bsma.pdbj.org/entry/26. Full trajectories for all replicas used in this research are available upon request.

In addition to these extended ensemble simulations, we performed 8 MD simulations of 500 ns length each, starting from the initial configuration of the nsp1-SL1 complex to see the difference between conventional simulations and extended ensemble simulations. First 50 ns chunks of the simulations were removed from the data as the equilibration time, and remaining 450 ns simulation results were used in the subsequent analyses.

## Simulation analysis

As the simulations were performed with modified potential functions, we performed the reweighting procedure to subtract the effect of modified potential functions. We used the multistate Bennett acceptance ratio (MBAR) method [41, 42] to calculate statistical weights to the structures in trajectories; in essence, structures that are difficult to obtain without potential modification have smaller weight values. With that method, we obtained a weighted ensemble corresponding to the canonical ensemble (trajectory with a weight assigned on each frame) from multiple simulations performed with different potentials. Only eight replicas corresponding to the eight lowest replica indices (i.e., the one with the unscaled potential function and seven replicas with the potentials closest to the unscaled potential) were used in the MBAR analysis. The weighted ensemble of the trajectory was used in the subsequent analyses. Visualization was performed using VMD [43] and pymol [44]. The secondary structure of nsp1 was analyzed using the definition of DSSP [45] with `mdtraj`. [46]

For both nsp1 simulations with and without SL1, we analyzed the intra-residue contact within nsp1. We defined the contact between nsp1 residues by having at least one inter-atomic distance between heavy atoms less than or equal to 4 Å, or having a residue number difference no larger than 2. From the simulation ensemble, we calculated the ratio of the contact between residues by taking the weighted average.

The relative orientation of the SL1 and nsp1 IDR was analyzed using principal axes (the easiest axis for the rotation) of two groups. For SL1, phosphate atoms in the stem loop region (residues 7–33) were used for the principal axis calculation; the sign of the principal axis vector

was chosen to match the direction along C19 to G7's phosphate atoms. For nsp1, C-terminal end of globular region and N-terminal side of IDR (residues 121–146) were used, and the sign was chosen such that the direction matches that from residue 121 C$\alpha$ atom to residue 146 C$\alpha$ atom. The angle between two axes was used to analyze the orientational preference between the two.

**Clustering.** After we obtained multiple poses of the nsp1-SL1 complex from the simulation, we classified structures into clusters. Typically, measures such as the root-mean-square deviation (RMSD) are used to distinguish different structures; however, for the IDR, the RMSD is not informative because the structures are more diverse, and also because the RMSD is extremely sensitive to motions far from the center of mass. We thus used the contact information between nsp1 and SL1 residues to analyze the simulation results. Here, inter-residue contacts were detected with the criterion that the inter-atomic distance between the C$\alpha$ of an amino acid residue and C4' of a nucleotide residue was less than or equal to 12 Å.

On the basis of the inter-residue contact information, the binding modes of the nsp1–SL1 complex observed in the ensemble were evaluated by applying the clustering method. The inter-residue contact information in each snapshot was represented as a contact map consisting of a $180 \times 36$ binary matrix. The distance between two snapshots was then calculated as the Euclidian distance of binary vectors with $180 \times 36 = 6480$ elements. We applied the DBSCAN method [47] to classify the binding modes. We arbitrarily determined two parameters, `eps` and `minPts`, for the DBSCAN method to obtain a reasonable number of clusters each of which had distinct binding modes (the validity of parameters is also discussed in Fig A in S1 Text). Note that the DBSCAN generates clusters each of which has more than `minPts` members based on the similarity threshold `eps`. The clusters with fewer than `minPts` members (including singletons) were treated as outliers. We used `eps = 6` and `minPts = 200` in this research.

After clusters were obtained, we applied two other criteria to characterize interactions between nsp1 and SL1 in each cluster. (i) Hydrogen bonds were detected with the criteria that the hydrogen-acceptor distance was less than 2.5 Å and the donor–hydrogen–acceptor angle was greater than 120 degrees. (ii) Salt-bridges were detected with the criterion that the distance between a phosphorous atom in the RNA backbone and the distal nitrogen atom of Arg or Lys was less than 4.0 Å.

We also extracted 10 representative structures corresponding to each cluster. These representative structures are also deposited to the BSMA archive. We assessed the stability of these representative structures in clusters 1 and 2 (clusters having the two largest populations) by running a simulation from representative structures. Four structures were sampled from cluster 1 and 2 each by randomly resampling structures with weight factors obtained from the reweighting. Then, 500 ns conventional MD simulations from these $4 \times 2$ structures were performed with new random initial velocities assigned. Resulting trajectories were converted to the binary matrix by the same procedure we used in the clustering, then the distances from the centers of clusters were calculated to assess the stability.

## Modeling nsp1–40*S* ribosome complex

To compare the binding poses obtained from the simulation with the recent experimental results, we modeled the complex structure of nsp1 and 40*S* ribosome based on the density map from the cryo-EM and the cross-linking experiment. We first modeled an nsp1–40S ribosome complex by the density fitting approach. It has been reported that, in the ribosome–nsp1 complex cryo-EM density map (Electron Microscopy Data Bank ID: EMD-11276), where a chunk of electron density was observed near the C-terminal structures of nsp1, which is considered

to be the N-terminal globular region of nsp1. [11] We fitted the SARS-CoV-2 nsp1 N-terminal domain structure (PDB ID: 7K3N) into the density map using the structure of 40S ribosome–nsp1 C-terminal helices complex (PDB ID: 6ZLW) to find appropriate candidates of nsp1 N-terminal region. We used UCSF Chimera [48] to fit the density map. Six models with the correlation coefficient greater than 0.80 were found, and were used for further analysis.

It has been reported that nsp1 and ribosomal protein S3 could form cross-links with targeted *in situ* cross-linking mass spectrometry. [49] Two inter-residue crosslinks between nsp1 K120–S3 K62 and nsp1 K141–S3 K108 were reported, where the lysine residue in nsp1 in the latter pair was mapped to the IDR. We measured the distance between C$\alpha$ atoms at nsp1 K120 and S3 K62 of 6 nsp1–40$S$ ribosome candidate structures. We selected candidate 2 as the model because the distance between cross-linked residues met the criterion ($< 25$ Å) and the number of collisions between C$\alpha$ atoms was the lowest (Table A in S1 Text). For the convenience of the readers we deposited the model structure of nsp1 bound to the ribosome to BSMA.

## Results and discussion

### Convergence of the extended ensemble simulation

We first monitored the convergence of the ensemble using the secondary structure distribution and the stability of the hydrogen bonds between nsp1 and SL1 (Text A and Figs B and C in S1 Text). The hydrogen bond and secondary structure statistics reached a plateau at $\sim 30$ ns. However, as expected from the relatively short simulation length and large number of replicas, the replica states were not well mixed. The replica state indices of each continuous trajectory were limited in a narrow range, demonstrating that the sampling is still insufficient (Fig D in S1 Text). Our simulation trajectories henceforth should be recognized as a set of meta-stable structures without acheiving the total convergence to the canonical ensemble. Nevertheless, the cluster analysis of conventional MD results starting from the initial structure indicates that the structures from REST2 extended ensemble simulations resulted in a totally different structure obtained in the conventional MD (Fig E in S1 Text). Furthermore, we observed that major structure clusters obtained from the REST2 simulation were stable with the conventional MD (we will discuss in "Clustering analysis of the binding poses"). The limitations of the present calculation will be discussed in "Limitations of this study".

### The IDR partially forms secondary structure and binds to SL1

Although we did not restrain the RNA-nsp1 distance in the simulation and started the simulation with the two molecules apart, they formed a complex within the simulation. Fig 1B shows a representative snapshot of the complex at the end of the simulation. The RNA stem binds to the C-terminal disordered region. However, as shown in Fig 1C, when the N-terminal domain of nsp1 was superimposed, the RNA structures did not have a specific conformation. This implies that there was no distinct, rigid structure mediating nsp1-RNA binding.

We next investigated the secondary structure of the nsp1 region simulated with SL1 (Fig 2). Although we started the simulation from an extended configuration, the C-terminal region at residues 153–179 partially formed two $\alpha$-helices, which is consistent with the fact that the C-terminal region forms two helices (residues 153–160, 166–179) and shuts down translation by capping the pore that mRNA goes through in the cryo-EM structural analysis. The result also indicates that the cap structure may be formed before nsp1 binds to the ribosome, reflecting a pre-existing equilibrium, although the ratio of the helix-forming structures is only up to 50%. In addition to these known helices, residues 140–150 also weakly formed a mixture of $\alpha$-helix and 3–10 helix. Residues at other regions (1–11, 128–139) remained disordered. We also

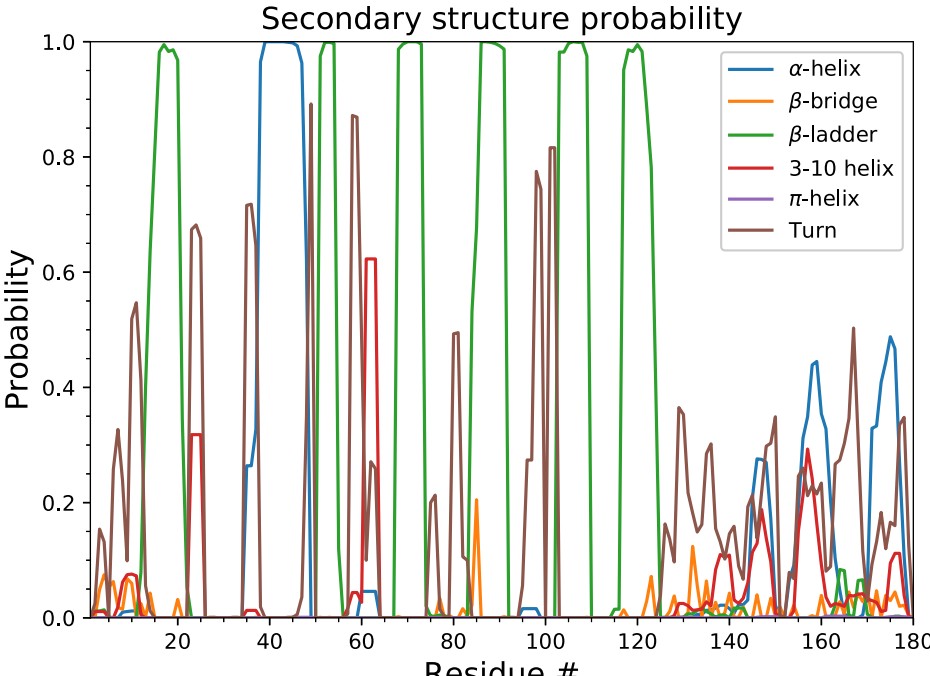

**Fig 2. Secondary structure distribution of nsp1.** Probabilities were calculated using the reweighting of the last 25 ns simulation trajectories.

**Table 1. Hydrogen bonds observed between SL1 and nsp1.**

| Nsp1 residue | Main/side | SL1 base | BB/base | % |
|---|---|---|---|---|
| Arg124 | Side | U18 | Backbone | 26.0 |
| Lys47 | Side | C16 | Backbone | 23.0 |
| Arg43 | Side | U17 | Backbone | 19.6 |
| Asn126 | Side | U17 | Backbone | 18.7 |
| Gly127 | Main | U18 | Backbone | 18.2 |
| Asn126 | Side | C20 | Base | 17.4 |
| Ser135 | Main | C20 | Base | 14.8 |
| Arg124 | Main | U17 | Base | 14.4 |
| Asn126 | Side | C20 | Backbone | 13.4 |
| Ser40 | Side | U17 | Backbone | 13.1 |
| Asn126 | Side | C16 | Backbone | 13.0 |
| Asp75 | Main | U18 | Base | 12.7 |
| Asn126 | Side | U18 | Backbone | 12.3 |
| Ala131 | Main | C19 | Base | 12.2 |
| Ser135 | Side | C16 | Sugar | 12.2 |
| Lys47 | Side | C20 | Backbone | 12.0 |
| Tyr136 | Main | C20 | Base | 11.9 |
| Ser135 | Side | C20 | Base | 11.6 |
| His134 | Main | C19 | Base | 10.8 |
| Asp75 | Side | U18 | Base | 10.4 |

investigated the structure of nsp1 without SL1. There were no substantial change in the secondary structures except slightly lower $\alpha$-helix formation ratio at residues 153–160 (Fig F in S1 Text).

### SL1's hairpin region binds to the nsp1 IDR

Inter-residue contact probabilities between nsp1 and SL1 in the canonical ensemble are summarized in Table 1 and Figs 3 and 4. Based on the distribution of the interactions, we categorized the binding interface of nsp1 into five regions (Fig 1D and Table A in S1 Text): (i) the N-terminus (residues 1–18), (ii) the $\alpha$1 helix (residues 31–50), (iii) the disordered loop between $\beta$3 and $\beta$4 (residues 74–90), (iv) C-terminal end of the globular region and the N-terminal side of the IDR (residues 121–146), and (v) the C-terminal side of the IDR (residues 147–180). These five regions interacted primarily with bases around C20 of the RNA fragment, which composes the stem loop. The most important region for recognition of SL1 was region (iv), the N-terminal side of the IDR. The probability of contacts between any residue in this region and SL1 was 97.4%. In particular, contact between Asn126 and U18 was observed in 84.1% of the canonical ensemble. The most frequently observed hydrogen bond in the canonical ensemble was Arg124–U18, the probability of which was 26.0% (Table 1). The second most important interface region was region (ii), $\alpha$1 helix, which has two basic residues (Arg43 and Lys47), that frequently formed salt-bridges with the backbone of SL1. At least one salt-bridge in this region was included in 69.8% of the canonical ensemble. The third most important was region (iii), consisting of the loop between $\beta$3 and $\beta$4; 63.2% of the canonical ensemble included at least one contact in this region. Asp75 sometimes formed hydrogen bonds with the bases of SL1.

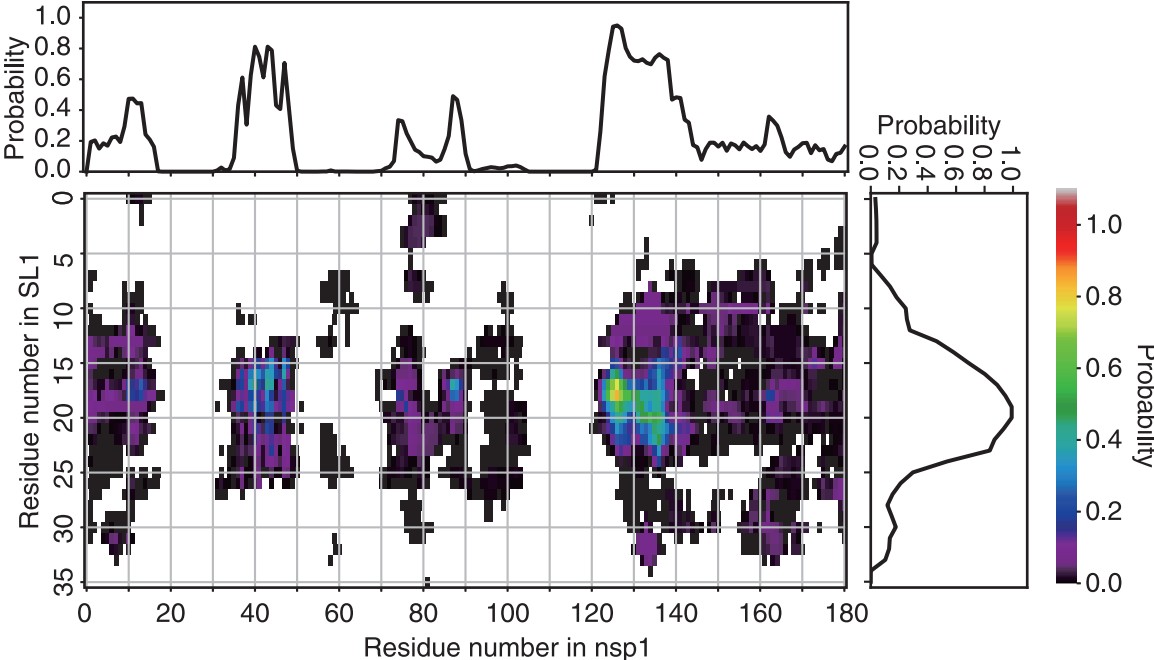

**Fig 3. Contact probabilities between nsp1 and RNA.** Residue-wise, all-against-all contact probability in the canonical ensemble. The color at each grid point indicates the statistical weight of the contact between the corresponding pair of residues (color scale is shown at the right of the panel). The points filled by white indicate no detectable probability of contacts. The line plots at the top and right of the contact map depict the contact probability for each residue, regardless of its counterpart.

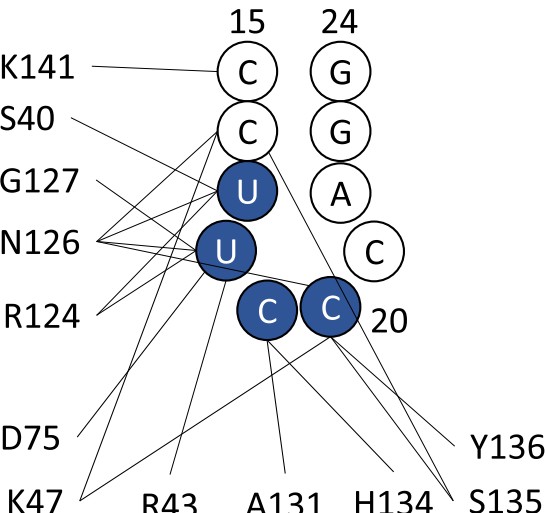

**Fig 4. Graphical representation of the hydrogen bond interactions between SL1 and nsp1.** Bases of U17 to C20 (colored blue) are recognized by the hydrogen bonds.

Regions (i) and (v) tended not to form hydrogen bonds or salt-bridges, but frequently contacted residues in these regions; the probability for interactions with regions (i) and (v) were 72.1% and 59.2%, respectively.

As an overall shape, the nsp1 surface consists of positive and negative electrostatic surface patches separated by a neutral region (Fig 5A). [50] The $\alpha$1 helix in region (i) forms the interface between these two patches; one side of the helix contains basic residues (Arg43 and Lys47), and the other side contains some hydrophobic residues (Val38, Leu39, Ala42, and Leu46). The positive side of the $\alpha$1 helix assumes a mound-like shape with a positively charged cliff (Fig 5B). The bottom of the valley formed by the N-terminus and $\beta$3-$\beta$4 loop, or regions (i) and (iii), respectively, also contains positive electrostatic potentials. The positively charged cliff and valley attract and fit to the negatively charged backbone of SL1. Eventually the IDRs in region (iv) and (v) grab SL1.

Although the binding site for SL1 on nsp1 can be characterized as an interface consisting of regions (i) through (v), SL1 did not assume a stable conformation, even when it was bound to these regions. Diverse binding modes were observed in the canonical ensemble. Although SL1 nearly always interacted with residues in the region (iv), its conformation was diverse and fluctuated greatly. In addition, the nsp1 IDR was also highly flexible.

## Nsp1's globular region and IDR do not stably interact with each other

Next, we investigated the intra-residue contacts within nsp1 with and without SL1. Fig 6 shows the contact map between nsp1 residues. The result shows that nsp1's globular region and IDR did not have stable contacts regardless of the presence of the SL1. We further analyzed the difference between two contact maps to investigate the specific changes in the structures (Fig 6 right). Overall, the difference in contacts was small, and thus nsp1 alone may not experience significant structural changes with and without SL1, which is consistent to the secondary structure analysis. The largest difference in the contact ratio appeared between residues Glu65 and Tyr68, which are located in the loop between $\alpha$2 helix and $\beta$3 helix. However, the ratio of the contacts between the loop on residues 64–68 and SL1 was low in the inter-contact

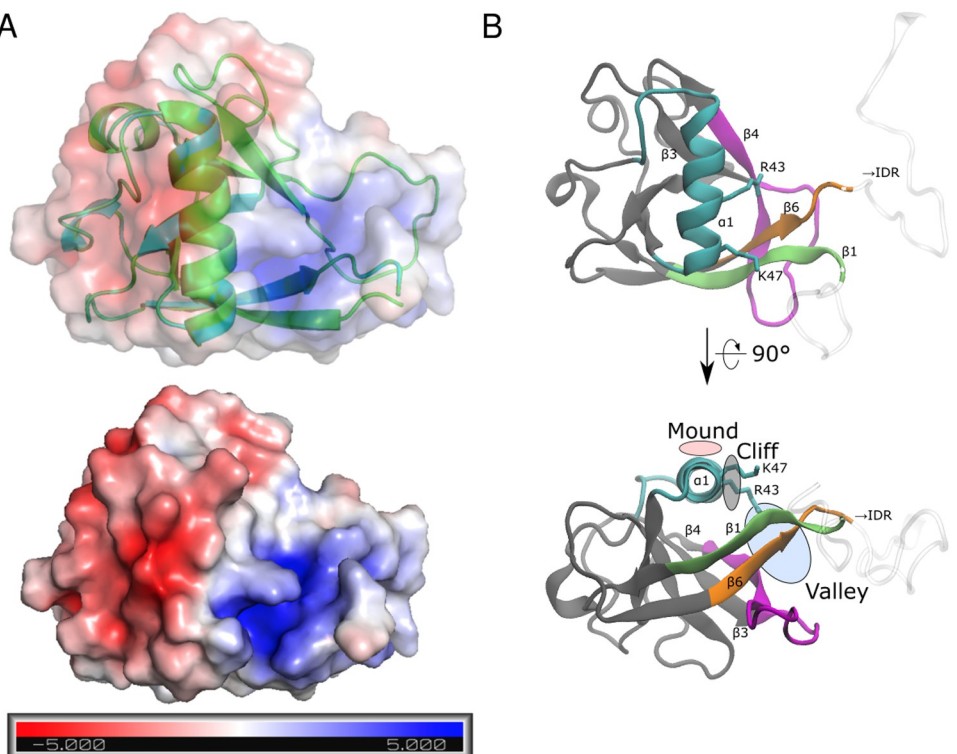

**Fig 5. Binding surface of nsp1.** (A) Surface electrostatic potential of the nsp1 and (B) annotated surface structure of the nsp1 recognition sites for SL1. In (A), units are in $k_B T/e$, where $k_B$ is the Boltzmann factor, $T$ is the temperature of the system (= 300 K), and $e$ is the unit charge of a proton. Color coding in (B) corresponds to the region defined in Fig 1D.

analysis (Fig 3), suggesting that the change in the loop structure is caused indirectly. Because there are also contact ratio changes at Gly30–Glu65 and Gly30–Gln66, and Gly30 is located next to $\alpha$1 helix, it is possible that the contact of $\alpha$1 to SL1 shifted $\alpha$1 and led to Glu65–Tyr68 contact difference.

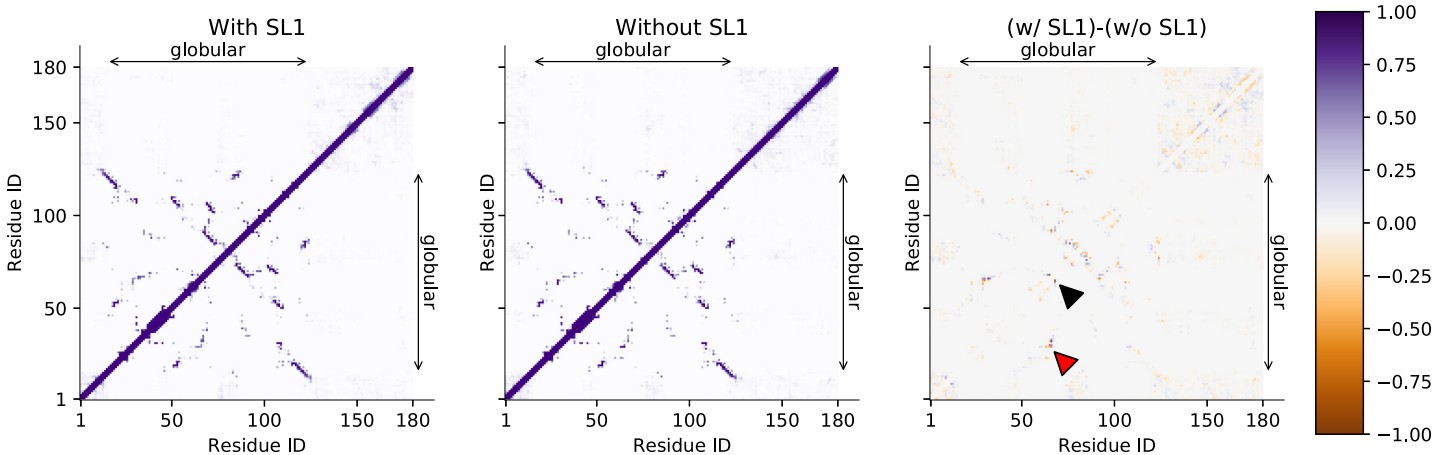

**Fig 6. Interactions within nsp1.** Contact probabilities between nsp1 residues were color-coded. Regions corresponding to residues 14–125, which are visible in the X-ray crystallographic structure (PDB ID 2HSX), are shown as arrows on the right and the top of each image. (Left, middle) the probabilities for nsp1 with and without SL1. (Right) the difference of the two (with SL1 minus without SL1). Residue pairs referenced in the main text are annotated by black and red wedges (pointing residue pairs 65–68 and 30–65, respectively).

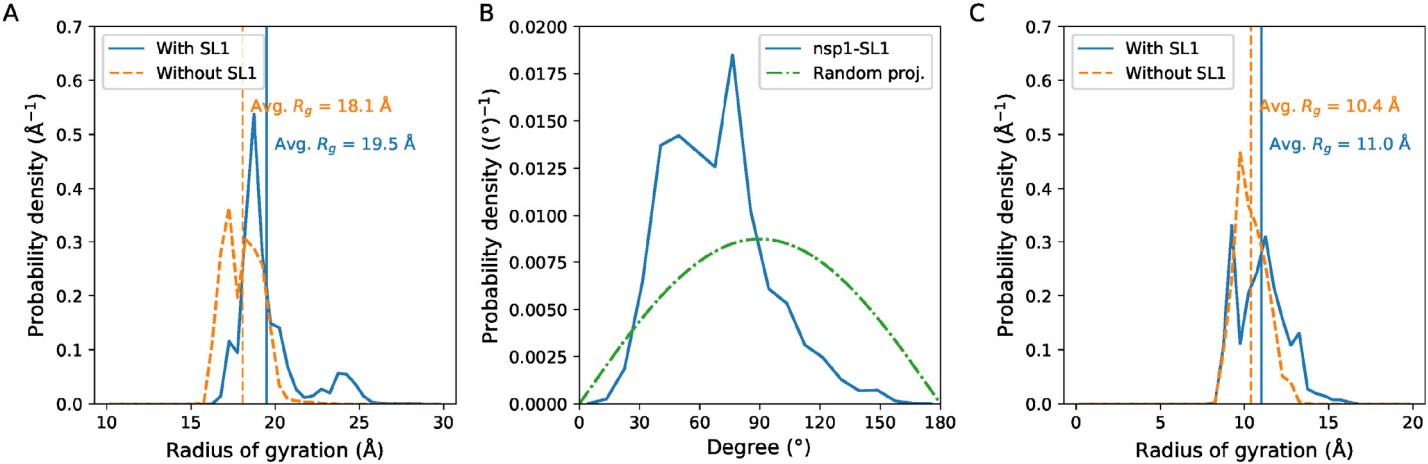

**Fig 7. Nsp1 IDR and SL1 are partially aligned.** (A) The radius of the gyration of nsp1 Cα atoms with or without SL1. Vertical lines represent the average. (B) The distribution of angle between SL1 and nsp1 residues 121–146 (see Materials and methods for the definition). The theoretical angle distribution of two random vectors are also presented for comparison. (C) The radius of the gyration of nsp1 C-terminal IDR. For (A)-(C), the ordinate represent the probability density, i.e. the ordinate are scaled so that the area under the curve is exactly 1.

## Nsp1's IDR may be aligned with SL1

Although there were no stable contacts between nsp1 globular region and IDR, the radius of gyration $R_g$ of the nsp1 without SL1 was smaller compared to that without SL1 (Fig 7A), suggesting that nsp1 alone is more compact compared to nsp1 with SL1. This result raises another question: why nsp1 is elongated under the presence of SL1 while having no interaction between nsp1 globular and IDR regions? We hypothesized that nsp1 is extended alongside the stable stem loop structure. We analyzed the angle between SL1 and nsp1 region (iv) (Fig 7B). The result indicated that the angle between the two was more likely to be < 90 deg, i.e., two axes were weakly aligned. As a result, the radius of the gyration of nsp1 region (iv) with SL1 was also larger than that without SL1 (Fig 7C).

## SL1's binding position in 40S ribosome-nsp1 complex

We further investigated the consistency between the known structure and the SL1's binding preference. For that purpose, we constructed the model of the 40S ribosome-nsp1 complex. Fig 8A shows the overall structure of the 40S ribosome-nsp1 complex and Fig 8B presents the closeup view around nsp1. The "valley" of nsp1 was close to the nsp1–S3 binding interface, albeit open to the solvent. Thus, SL1 has enough space for binding even in the presence of the 40S ribosome. These results suggest that SL1 may form the trimer complex with the ribosome and the nsp1.

We note that in addition to the reported interactions between nsp1 and S3, the C-terminal disordered region of ribosomal protein S10 is also in proximity to nsp1 and the putative binding site of SL1 in the complex structure. The result suggests that nsp1 and/or SL1 may have interactions with the disordered C-terminal tail of S10.

## Clustering analysis of the binding poses

The diversity of the binding modes was further investigated using cluster analysis based on the contact map for each snapshot (see Materials and methods). We determined the clustering threshold using the criterion that any cluster has at least one inter-residue contacts with more than 80% in each cluster. As a result, the binding modes could be categorized into 14 clusters

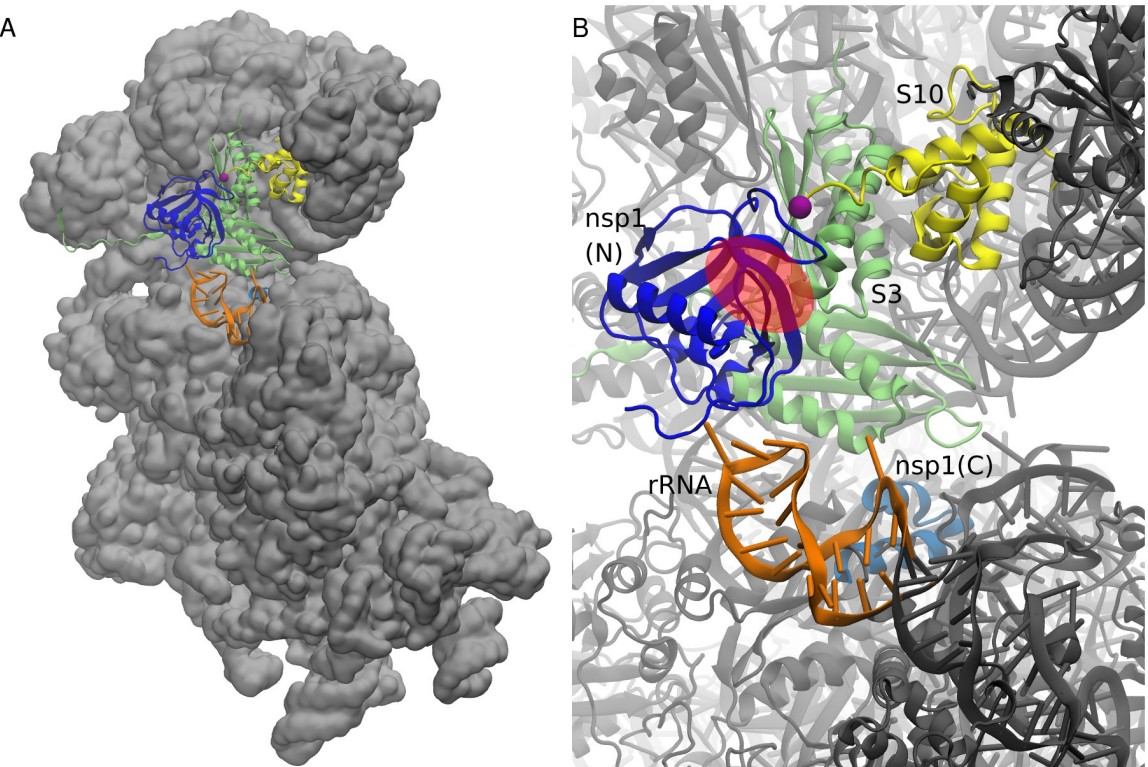

**Fig 8. Mapping of the binding surface over the nsp1–40S ribosome complex.** (A) Overview of the nsp1–40S ribosome complex structure modeled from cryo-EM structure and its electron density map combined with the cross-linking mass spectrometry. (B) Close-up view of the structures around nsp1. The N-terminal and C-terminal parts of nsp1 are colored blue and cyan. A hairpin of rRNA (residue number 531 to 550) is colored orange, and the ribosomal proteins S3 and S10 are colored yellow and lime green, respectively. C-terminal region of S10 after residue 97 (purple sphere) is considered to be the disordered region and is not visible in the structure. Region corresponding to the "valley" of nsp1 binding surface is presented as a red transparent circle.

and outliers, which had 34.2% of the statistical weight in the canonical ensemble. In even the most major cluster, the statistical weight was only 15.5%; those for the second, third, and fourth clusters were 9.9%, 7.4%, and 5.0%, respectively. Each cluster had a unique tendency to use a set of binding regions (Text C and Fig G in S1 Text). We also analyzed the differences in surface areas of the interacting interfaces in the ordered and disordered regions of nsp1 among the 14 clusters (Fig H in S1 Text). The distribution shows the unique characteristics of each cluster. These results indicate that SL1 binds to nsp1 by multimodal binding modes.

The representative structure of cluster 1, which had the largest population among all clusters, is presented in Fig 9 and Table C in S1 Text. Nsp1 recognized SL1 via regions (ii), (iii) and (iv). In the region (ii), the basic residues in H2 formed the Arg43–C17 and Lys47–U16 salt-bridges. Region (iii) recognized SL1 via the Asp75–U18 hydrogen bond. Residues Arg124 through Gly137 in region (iv) attached to SL1 via the Arg124–U17, Ala131–C19, and Ser135–C16 hydrogen bonds; Tyr136 stacks between C21 and G23 instead of A22, which was flipped out. Representative structures of clusters 2 and 3 are also presented in the supporting material (Text C, Figs I and J, and Tables C and D in S1 Text).

The stability of the obtained structures in the cluster was assessed with the conventional MD simulation. Starting from 8 structures of clusters 1 and 2 (4 structures each), we performed 500 ns MD simulations (4 μs in total) and analyzed whether the structure stably maintains the configuration found in the simulation. Fig K in S1 Text presents the nsp1-SL1

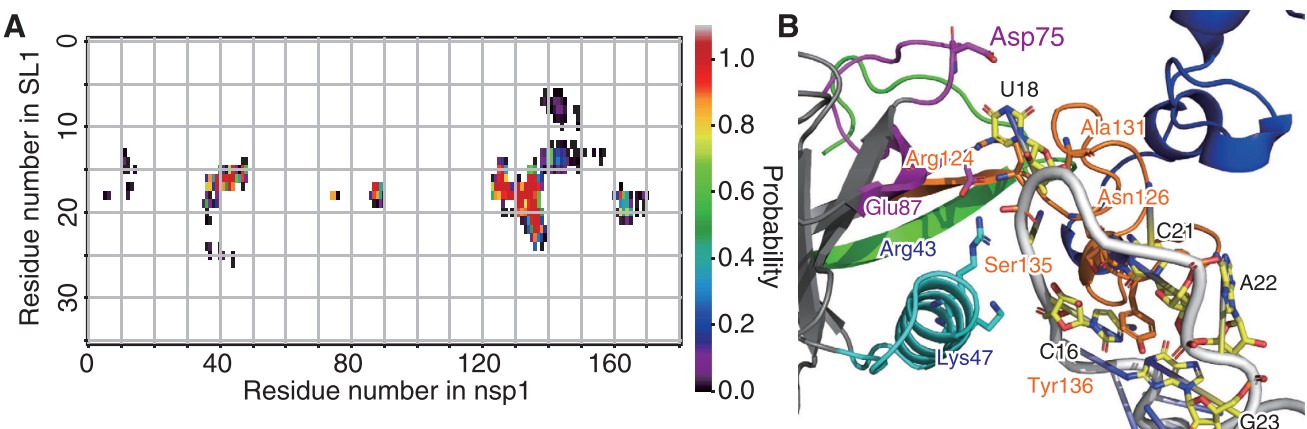

**Fig 9. Interactions between nsp1 and SL1 observed in cluster 1.** (A) Pairwise contact probability in cluster 1. See the legend to Fig 3. (B) Representative snapshot of the cluster 1. The interface regions (i) through (v) are shown as green, cyan, magenta, red, and blue ribbons. Bases 16–26 of SL1 are shown in orange.

contact map distances between the trajectory of conventional MD simulations and the cluster center. All the four trajectories started from the most populated cluster (cluster 1) kept their conformations during 500 ns simulations. The simulations started from the second most populated cluster (cluster 2) were less stable. Two trajectories showed conformational changes around 200 ns while the other two kept their conformations. Therefore, the structures found in the cluster analysis, especially cluster 1, are considered stable for a reasonable time span.

## Relation to other experimental results

It has been reported that the Arg124Ala–Lys125Ala double nsp1 mutant lacks the ability to recognize viral RNA. [3, 51] This can be explained by the results of our simulation, which showed that sidechain of Arg124 strongly interacts with the phosphate backbone of U18 (Table 1 and Figs 4 and 9). An Arg124Ala mutation would eliminate the ionic interaction between the sidechain and the backbone, and nsp1 would lose its ability to recognize viral RNA. Additionally, Arg124 and Lys125 are not contacting to the ribosome in the model structure, which is consistent to the fact that the UV cross-linking to 18S RNA was unaffected by these two mutations. [15] On the other hand, recently reported Arg99Ala mutation to nsp1, which also lacks the ability to recognize viral RNA, [51] did not match important hydrogen bonds we found in the top clusters. This may be attributed to the insufficient sampling around Arg99 (it is not included in the REST2 region) and/or lack of important binding partners in the system, e.g. the ribosome.

The circular dichroism spectrum of the SARS-CoV-2 nsp1 C-terminal region (residues 130–180) [17] in solution had only a single peak at 198 nm and did not show ellipticities at 208 nm and 222 nm. This indicates that the nsp1 C-terminal region did not form $\alpha$-helices or $\beta$-sheets and was disordered. Similarly, in the analysis of NMR [52] spectra, nsp1 N- and C-termini are predicted to be fully unstructuted, but the predicted order parameters were different among residues. Notably, relatively low order parameters were observed for residues 165–180 (corresponds to the $\alpha$-helix region that shuts the ribosome), which is inconsistent with our simulation result. Although in our simulation we found that nsp1 partially forms the $\alpha$-helix in the IDR, our simulation also showed that the percentage of the helix in the IDR was low (<60%) and the structure was unstable, which may explain the difference from the experimental results; without SL1 the propensity was lowered further (Fig 2, and Fig F in S1 Text). Note

that these experiments were conducted without SL1. The propensity of the structure formation may be affected in the presence of the highly charged molecules like RNA. Further study will be needed before a conclusion can be drawn.

In X-ray crystallographic analysis of SARS-CoV-2 nsp1 N-terminal region, [50] $\beta$5-strand of residues 95–97 only exits in SARS-CoV-2 nsp1 and not in SARS-CoV, despite the sequences at residues 95–97 were unchanged. It was thus considered as a characteristic difference between the two, although the site was near the crystal contact. In NMR, however, $\beta$5-strand was not observed, [52] which was also corroborated by the order parameter and NOE analysis. Our simulation data supports the latter, where residues 95–97 did not form $\beta$-ladder as shown in Fig 2.

Whether SARS-CoV-2 nsp1 and SL1 bind without the ribosome is controversial. It has been reported that nsp1 and bases 7–33 of SARS-CoV-2 bind with a binding constant of 0.18 μM [53], but it has also been reported that a gel shift does not occur with the 5'-UTR of SARS-CoV-2 at concentrations up to 20 μM when tRNAs was used to exclude the non-specific binding. [6] The present simulation results indicate that the binding mode observed herein did not have a specific, defined structure. Typically, with such binding modes, the binding is expected to be weak. Therefore, these simulation results do not contradict with the results from either of the aforementioned experiments.

Mutations to SL1 bases 14–25, which disrupt the Watson-Crick pairs of the stem loop, reportedly cause translation to be shut off. [6] That observation is consistent with our finding that the hairpin structure of bases 18–22 in SL1 is recognized by nsp1. Hydrogen-bond interaction analysis showed that the RNA phosphate backbone is mainly recognized within the C15-C20 region (Table 1 and Fig 4). Moreover, our finding is consistent with the fact that the sequence of the hairpin region (corresponding to U18-C21 in our simulation) is not well conserved among SARS-CoV-2 mutational variants, whereas that of the stem is well conserved. [54] Our simulation shows that the interaction between nsp1 and the SL1 backbone is stronger than that between nsp1 and the SL1 sidechains (Table 1), which highlights the importance of the backbone interaction.

## Limitations of this study

Our simulations were performed based on several assumptions. Here, we list the limitations of the present study.

First, as we explained in "Convergence of the extended ensemble simulation", even though the current simulation uses the extended ensemble method, it is difficult to achieve full convergence. Sampling RNA structures are generally considered difficult even with the small system size, [55–59] and so does sampling the protein-RNA interaction. Given the length of IDR and the size of RNAs, the convergence of the simulation may be beyond the capability of the current computational resources. Current simulation results should thus be considered to achieve only partial convergence at best, i.e., current structures may not be the fully determined most stable structure under the current simulation force field, nor may it encounter enough transitions to obtain unbiased samples. [60] Therefore, in this research, we avoided the quantitative discussion of the energetics, which require complete convergence of the simulation; furthermore, the structures obtained in this research should be treated with caution.

Our simulations were performed without the ribosome. This was mainly because the simulation started before the structure of the nsp1-ribosome complex as well as the cross-linking experiment results were deposited. Furthermore, with the 40$S$ ribosome, ribosomal proteins S3 and S10 as well as rRNA hairpins at around residue 540 may interact with nsp1 or the SL1 as presented in Fig 8, which makes a proper sampling of the configurations difficult. With the

40*S* ribosome, the environment around nsp1 may be altered and so be the interaction between the RNA and nsp1.

To maintain the stability of the hairpin loop structure, we performed the simulation with restraints on the G-C pairs in the 5'-UTR. These restraints may have hindered RNA forming structures other than the initial hairpin structure. However, in the secondary structure prediction using CentroidFold [61] and the reference sequence, these base pairs were predicted to exist in more than 92% of the ensemble. Furthermore, a recent study [59] showed that, even with a rigorous extended ensemble simulation, the hairpin structure remained intact. Given these results, the drawback of structural restraints to SL1 is expected to be minimal.

Finally, as is always the case with a simulation study, the mismatch between the simulation force field and the real world leaves a non-negligible gap. For the simulation of IDR, AMBER14SB used in this research may favor the folded state. [62] To overcome this problem, several force fields specialized for IDR simulations have been proposed. [63, 64] However, IDR-oriented force fields are not suitable to simulate ordered regions in general, and are not always better than conventional force fields even in IDR simulations. [65, 66] In this study we used AMBER14SB for proteins to balance the stability of both globular and disordered regions. The result may depend on the force field used, e.g., the high propensity of the folded state on the C-terminal region may be attributed to the property of the force field. Not only force fields for proteins, but the choices for RNA force fields should also be considered, as each force field has different characteristics upon reproducing RNA structures as well as protein-RNA complexes [57, 58, 67]. The simulations with multiple different force fields will be almost necessary to avoid drawing conclusions biased by a specific force-field. In addition to the force field issues, some residues may have alternative protonation states upon binding to RNA (e.g., histidine protonation state), which should be investigated further.

## Conclusion

### Future research and conclusions

The present simulation was performed with only nsp1 and SL1. Arguably, simulation of a complex consisting of the 40*S* ribosome, nsp1 and SL1 will be an important step toward further understanding the details of the mechanism underlying the evasion of nsp1 by viral RNA. Our results suggest that the nsp1-SL1 complex without ribosome has multimodal binding structures. The addition of the 40*S* ribosome to the system may restrict the structure to a smaller number of possible binding poses and possibly tighter binding poses may be obtained, while the convergence of the simulation may be mitigated. However, as shown in Fig 8B, in addition to the contacts between nsp1 and S3 and rRNA around residue 540, the C-terminal IDR of S10 may also interfere with nsp1, which may make sampling proper configurations more difficult. Additionally, recent researches suggest possible caveats and remedies in the REST2 protocol; [68, 69] the combination of methodological advances and more refined models may enable us to sample structures such that the stability of the complex can be discussed quantitatively. Further researches will be necessary in this direction.

In addition to a simulation study, mutational analysis of nsp1 will be informative. In addition to the already known mutation at Arg124, current simulation results predict Lys47, Arg43, and Asn126 are important to nsp1-SL1 bninding. Mutation analyses of these residues will help us to understand the molecular mechanism of nsp1.

Finally, the development of inhibitors of nsp1-stem loop binding, is highly anticipated in the current pandemic. Although the present results imply that a specific binding structure might not exist, important residues in nsp1 and bases in SL1 were detected. Blocking or mimicking the binding of these residues/bases, could potentially nullify the function of nsp1.

In conclusion, using MD simulation, we investigated the binding and molecular mechanism of SARS-CoV-2 nsp1 and the 5'-UTR stem loop of SARS-CoV-2 RNA. The results suggest that the 5'-UTR stem loop of SARS-CoV-2 has the preference of binding onto regions spanned from $\alpha$1 helix to the disordered region. Upon the binding, the disordered region may extend along the stem loop. The interaction analysis further suggested that the hairpin loop structure of the 5'-UTR stem loop binds to the N-terminal domain and the intrinsically disordered region of nsp1. Combined with the modeling, in the presence of the ribosome, the 5'-UTR stem loop may bind to the interface of nsp1 and ribosomal protein S3, and ribosomal protein S10 may also be involved in recognition of the 5'-UTR stem loop. Multiple binding poses of nsp1 and the stem loop were obtained, and the largest cluster of the binding poses included interactions that can explain the results of the cryo-EM, the cross-linking experiments, and the previous mutational analyses.

## Supporting information

**S1 Text. Supporting information document.** Text A: Convergence of the simulations. Text B: Characterstics of clusters. Text C: Details of clusters 2 and 3. Fig A: Survey for the clustering parameters. Fig B: Convergence of the secondary structure distribution. Fig C: Convergence of the hydrogen bond forming ratio. Fig D: Timecourse of the replica indices. Fig E: Distances to clusters in canonical simulations starting from the initial configuration. Fig F: Secondary structure distribution of nsp1 without SL1. Fig G: Representative structure of each cluster. Fig H: Surface area of interaction intefaces of nsp1. Fig I: Interactions between nsp1 and SL1 in cluster 2. Fig J: Interactions between nsp1 and SL1 in cluster 3. Fig K: Distances to clusters in canonical simulations starting from cluster 1 structures. Table A: Characteristics of the nsp1–40S ribosome complex models. Table B: Characteristics of SL1 binding regions of nsp1. Table C: Characteristics of each conformational cluster.
(PDF)

**S1 File. RNA force field file used in this work.**
(ZIP)

**S2 File. Patches applied to GROMACS 2016 used in this work.**
(ZIP)

## Acknowledgments

We thank Dr. Atsushi Matsumoto for his technical assistance. Simulations were performed on supercomputers at Research Center for Computational Science, Okazaki, and Academic Center for Computing and Media Studies, Kyoto University.

## Author Contributions

**Conceptualization:** Shun Sakuraba, Hidetoshi Kono.

**Formal analysis:** Shun Sakuraba, Qilin Xie, Kota Kasahara, Junichi Iwakiri.

**Funding acquisition:** Shun Sakuraba, Kota Kasahara, Junichi Iwakiri, Hidetoshi Kono.

**Methodology:** Shun Sakuraba, Kota Kasahara, Junichi Iwakiri.

**Project administration:** Shun Sakuraba.

**Resources:** Hidetoshi Kono.

**Software:** Shun Sakuraba.

**Supervision:** Kota Kasahara, Hidetoshi Kono.

**Visualization:** Shun Sakuraba, Kota Kasahara.

**Writing – original draft:** Shun Sakuraba, Qilin Xie, Kota Kasahara.

**Writing – review & editing:** Shun Sakuraba, Kota Kasahara, Junichi Iwakiri, Hidetoshi Kono.

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
