## [Decision Letter · Decision Letter 0]

2 Aug 2021

Dear Dr. Sakuraba,

Thank you very much for submitting your manuscript "Extended ensemble simulations of a SARS-CoV-2 nsp1-5'-UTR complex" for consideration at PLOS Computational Biology.

As with all papers reviewed by the journal, your manuscript was reviewed by members of the editorial board and by several independent reviewers. In light of the reviews (below this email), we would like to invite the resubmission of a significantly-revised version that carefully takes into account the comments of both reviewers.

We cannot make any decision about publication until we have seen the revised manuscript and your response to the reviewers' comments. Your revised manuscript is also likely to be sent to reviewers for further evaluation.

Sincerely,

Bert L. de Groot

Associate Editor

PLOS Computational Biology

Nir Ben-Tal

Deputy Editor

PLOS Computational Biology

Reviewer's Responses to Questions

**Comments to the Authors:**

Reviewer #1: Sakuraba et al present a very well written study about the mechanisms by which SARS-CoV2's cognate RNA evades the translation shutoff operated by viral protein nsp1 interacting with the host ribosome. Using enhanced sampling molecular dynamics simulations, they investigate the specific interaction between nsp1 and the SL1 region of the RNA. Needless to say, research on SARS-CoV2 is very valuable and timely. However, working on SARS-CoV2 implies both high rewards and high risks: the competition on this topic is fierce, and advances are published almost daily, rendering some papers "obsolete" even before they can be published. This is typically the case here.

The main issue is the the absence of the ribosome in the simulations: interactions between nsp1 and the ribosome are almost certain to change the conformation and/or conformational dynamics of nsp1, especially considering its partly unstructured nature, and no evidence exists that nsp1 binds the viral RNA in the absence of the ribosome ("Whether SARS-CoV-2 nsp1 and SL1 bind without the ribosome is controversial."). The authors are aware of this and very clearly state it in the "limitations of this study" section. It is unfortunate for the authors that the structure of the ribosome/nsp1 complex was published after their simulations had been performed and before this work could be published, but this does not change the fact that the mechanisms that the authors are exploring are, at best, tentative, and at worst, nonexistent.

To alleviate this, the authors should try to incorporate information from the nsp1/ribosome structure into their simulations and compare with their current results. This could be done for example by running additional REST2 simulations (possibly with less replicas than the original simulations) in which the ribosome-bound conformation of nsp1 is imposed using restraints, or by evaluating the difference in free energy between both states of nsp1... Thus providing information about the role of the ribosome in the recognition mechanism between nsp1 and SL1 would be quite valuable.

In any case, until additional simulations are performed, the paragraph "Distance between the nsp1 N-terminal domain and C-terminal helices" is much too tentative and should be removed. In my opinion, the assumption "This indicates that the configuration observed in the cryo-EM structure, which does not include SL1, is unlikely to happen when nsp1 is complexed with the SL1" cannot be safely made when one side of the comparison is done with the ribosome and the other one without.

Another important issue is the use of constraints on the RNA. In the "limitations" section, the authors claim that they are not really needed and thus do not impact the results meaningfully. But then, why were they used? What is the point of having SL1 in the "hot" region of the REST2 method if it is restrained? Figure 1C shows that no specific conformation emerges from the simulations for SL1; how is this compatible with the use of restraints, and what would happen without them? The authors should clarify this better than what is currently done in "limitations", and/or remove the restraints altogether in the additional simulations suggested above.

Another possible issue (this time not mentioned by the authors) relates to starting the MD simulations from an unbound state of the nsp1/RNA complex. Considering the length of the IDR in its extended conformation, differences in the initial positioning of the RNA along the IDR could result in different locations for the first contact between partners, conditioning further binding events (in particular the partial folding of the bound IDR that the authors observe). Yet the representativity of the binding simulation is not discussed by the authors at all...

A few more minor points:

- DBSCAN clustering: silhouette and/or Davies-Bouldin scores should be provided to assess the meaningfulness of the clustering and to justify the choice of the clustering hyperparameters;

- The right panel in figure 3 is not very informative: it is zoomed out too much, making the binding mode of nsp1 hard to grasp. I would suggest replacing the left panel by the right panel, and replacing the right panel by a zoomed-in view centered on the nsp1 binding site.

To conclude, reviewing this work is a bit of a conundrum for me. On the one hand, the article is well-written, the methods used are state-of-the-art, and the authors are quite straightforward about the limitations; on the other hand, it is debatable whether or not the results advance our actual understanding of SARS-CoV2's infection machinery. I hope additional simulations can be carried out, and major revisions to the article performed, to fortify the article's findings and make it suitable for publication in PLoS CB.

Reviewer #2: The work investigates, by MD simulations, a challenging system, SARS-Cov-2 nsp1 - 5’-UTR protein-RNA complex.

The paper is based on one REST2 enhanced sampling method atomistic simulation with impressive 192 replicas on one side but quite short 50 ns simulation time per replica on the other, resulting in 9.6 microsecond total time, which is not bad considering the size of the system. The currently recommended first-choice versions of the AMBER force field are used. Due to the absence of relevant structural data, preparation of the starting structures was very challenging. The authors have used a relevant (considering the circumstances) protocol. In summary, the work deserves publication. However, substantial revisions are needed before it can be published in order to better understand the significance/limitations of the computations, to make the paper more understandable for non-experts and to avoid overstating/over-interpretations.

Obviously, considering the inevitable approximations in the preparation of the starting structure, known deficiencies of both the protein and RNA force fields (which often struggle with even much simpler and better defined systems) and the obvious sampling limitations, a quantitative accuracy of simulations of the present system is not achievable. This of course does not preclude the publication, as I noted above. However, I request the authors moderate their tone, rather telling that the modelling “suggests certain things” than “reveals things”. The paper actually already contains a useful paragraph discussing limitations of the simulations, but the discussion should be extended by noting well-known (albeit often ignored) problems that can complicate simulations of protein-RNA complexes (https://wires.onlinelibrary.wiley.com/doi/10.1002/wrna.1405, https://pubs.acs.org/doi/10.1021/acs.chemrev.7b00427) and also sampling limitations which I assume are severe; see below.

The moderation would not undermine the paper while the reader would have a more balanced information. The credibility of the paper may actually be improved. There are currently many trashy MD papers in the literature which to my opinion degrade the MD simulation field. So, the authors may actually profit from distancing their work from that part of the literature, as their calculations per se are interesting.

The supplementary information presents very short discussion of the convergence of the REST2 simulation based on secondary structure and H-bond rate in the basic replica 0. To the best of my knowledge, it is not reliable analysis of a convergence of RE simulation so the claim of convergence is deceptive.

To analyze convergence would require to first monitor how do the continuous replicas travel in the replica space, which should be documented, and in addition convergence would have to be proven in the space of all continuous replicas. I do not think the simulations are converged, considering the complexity of the system. For such a system 50 ns RE simulation, even with 192 replicas, is insufficient to simulate converged binding. While more replicas may help reduce trapping, they reduce the time accumulated in the unbiased replica. Such simulations can maximally show some encounter complexes but would not reveal, to my opinion, true binding events in case the binding requires any significant induced fit mechanism for which the molecules would need to diffuse through the conformational space. For methods such as REST2 I can imagine realistically an order of magnitude speed up compared to plain simulations, but not more. Note also that efficiency of the RE methods is highly system-dependent and may sometimes be even compromised when higher-order replicas excessively increase the sampled space, when entropy and diffusive barriers are present etc. The folding/binding events are in reality finite-time single molecule events/micro-pathways which may prevent their realization in RE simulations (which obscure kinetics) for systems requiring more physical time to traverse the conformational space.

Enhanced sampling is not a panacea and rigorous analyses of performance often reveal problems https://doi.org/10.1146/annurev-biophys-042910-155255 Based on the presented data I have absolutely no idea about the convergence and I would assume that the REST2 run shows rather some mixture of encounter complexes accessible from the starting structure, not necessarily the ultimate binding. Thus, the section discussing limitations should be considerably broadened and the authors need to either monitor convergence by some valid tools or clearly admit that they did not monitor convergence at all. Some examples of papers with similar methods where convergence is better analyzed are for example: https://rnajournal.cshlp.org/content/21/9/1578.long, https://pubs.acs.org/doi/pdf/10.1021/acs.jctc.8b00955, https://www.nature.com/articles/s41467-021-21105-7; there are certainly also other works where convergence of RE simulations is analyzed rigorously. I emphasize that I do not expect the authors to prove the convergence of their ensemble as that is likely far beyond the currently available simulation techniques and hardware. Some degree of convergence can be achieved for fast folding proteins or very small nucleic acids system, or for mutual binding of essentially semi-rigid systems (say two folded proteins), but not for this system. However, discussion of the limitation and perhaps some insight how extended the sampling is would be essential to make the work understandable. Without this the analysis is rather hand-waving. I would tend to assume that the REST2 run reflects rather pre-binding of the complex within the approximation of the force field and assuming the start. Key parts of the binding landscape might still remain unvisited.

I would also suggest that authors briefly explain basics of the REST2 method and how it compares with other enhanced sampling methods. No mathematics is needed, just some plain explanation for non-experts, perhaps using SI if more space is needed. Presently the paper is a common MD simulation text with lot of technical jargon which is understandable for specialists but I do not think experimental researchers would grasp the computations. It is also one of the reasons why the impact of atomistic MD studies of protein-RNA complexes in the literature outside the internal MD community is not large. Other researchers often do not understand the papers and do not trust the methods due to notorious overstatements, over-interpretations and excessive self-confidence. Actually, could you please explain better which points exactly are used for clustering, what exactly is in the matrix?

To improve the paper, for comparison, it might be very useful to add to the paper a moderate set of standard simulations, first to see what happens when initiating from the starting structure in standard simulations and then perhaps initiating some simulations from interesting candidate structures detected in REST2. To see, if they settle down in some metastable conformations.

Interpretation of the paragraph “Distance between the nsp1 N-terminal…..”. Is it not possible that the short distance in the MD ensemble (if I understood the text correctly) is merely due to over-compaction of the protein by the force field? Unless simulations of the free protein are made for comparison, it is difficult to judge, to my opinion. Here even standard simulation of the protein can help (considering the timescale of the REST2 it looks that the effect occurs fast).

**Have the authors made all data and (if applicable) computational code underlying the findings in their manuscript fully available?**

Reviewer #1: Yes

Reviewer #2: None

PLOS authors have the option to publish the peer review history of their article (what does this mean?). If published, this will include your full peer review and any attached files.

Reviewer #1: No

Reviewer #2: No
---

## [Decision Letter · Decision Letter 1]

8 Dec 2021

Dear Dr. Sakuraba,

Thank you very much for submitting your manuscript "Extended ensemble simulations of a SARS-CoV-2 nsp1-5'-UTR complex" for consideration at PLOS Computational Biology. As with all papers reviewed by the journal, your manuscript was reviewed by members of the editorial board and by several independent reviewers. The reviewers appreciated the attention to an important topic. Based on the reviews, we are likely to accept this manuscript for publication, providing that you modify the manuscript according to the review recommendations of reviewer 2 in terms of more explicitly formulating the limitations of the current work in the manuscript.

Sincerely,

Bert L. de Groot

Associate Editor

PLOS Computational Biology

Nir Ben-Tal

Deputy Editor

PLOS Computational Biology

[LINK]

Reviewer's Responses to Questions

**Comments to the Authors:**

Reviewer #1: The authors have performed significant additional simulations regarding the stability of their nsp1/SL1 complex conformations, and their compatibility with the recently published nsp1/ribosome complex structure. They have thoroughly modified their manuscript accordingly, moderating the overall message and adding valuable new points to the discussion. My concerns have been addressed and I'm happy to recommend this new version of the manuscript for publication.

Reviewer #2: The revision has been done more or less appropriately, although the convergence issue is bigger than expected/admitted by the authors. On the other hand, amount of simulations is reasonably large in the context of contemporary literature and, mainly, the system is overwhelmingly complex. Thus, from my point of view the paper is OK for publication. (Obviously, I cannot speak for the other Reviewer who commented on several problems that are outside my expertise and which I cannot competently judge.) Still, I would tend to suggest as minor revision to add very few references at places where the authors discuss the limitations, which would (without any further explanations) guide interested readers to the literature where a more thorough discussion can be found.

p. 6. “However, as expected from the relatively short simulation length and large number of replicas, the replica states were not well mixed. The replica state indices of each continuous trajectory were limited in a narrow range,

showing a sign of insufficient sampling (Fig. SF4 in S1 Text)”.

It obviously is not a sign of insufficient sampling but a clear demonstration of it. However, with the present system one can hardly do more. Still, some readers might be unaware what is the significance/meaning of not being well mixed and further, might be unaware that even good mixing would not guarantee a convergence. There are other issues such as the lack of decorrelation, which may be problem even in case of at first sight good replica mixing in RE (the replicas can travel well across the ladder and still there could be lack of decorrelation). Simple cross-referencing without further explanations to e.g., the Zuckerman’s review https://www.annualreviews.org/doi/10.1146/annurev-biophys-042910-155255 plus the recent work https://www.nature.com/articles/s41467-021-21105-7 could help the readers. The former paper quite well explains the requirements for convergence: effective sample size, initial equilibration vs. correlation time etc. The latter showing system-dependent difficulties of specifically REST2.

The Figure SF4 is useful.

Similarly on p. 11 somewhere around line 460 probably a general reference to prot-RNA systems might be useful, e.g., https://pubs.acs.org/doi/10.1021/acs.chemrev.7b00427. There are specific layers of problems when simulating protein-RNA interactions compared to just RNA or protein simulations.

**Have the authors made all data and (if applicable) computational code underlying the findings in their manuscript fully available?**

Reviewer #1: Yes

Reviewer #2: None

PLOS authors have the option to publish the peer review history of their article (what does this mean?). If published, this will include your full peer review and any attached files.

Reviewer #1: No

Reviewer #2: No

Figure Files:

Data Requirements:

Reproducibility:

References:

---

## [Editor Report · Decision Letter 2]

4 Jan 2022

Dear Dr. Sakuraba,

We are pleased to inform you that your manuscript 'Extended ensemble simulations of a SARS-CoV-2 nsp1-5'-UTR complex' has been provisionally accepted for publication in PLOS Computational Biology.

Best regards,

Bert L. de Groot

Associate Editor

PLOS Computational Biology

Nir Ben-Tal

Deputy Editor

PLOS Computational Biology

---

## [Editor Report · Acceptance letter]

14 Jan 2022

PCOMPBIOL-D-21-01194R2 

Extended ensemble simulations of a SARS-CoV-2 nsp1-5'-UTR complex

Dear Dr Sakuraba,

I am pleased to inform you that your manuscript has been formally accepted for publication in PLOS Computational Biology. Your manuscript is now with our production department and you will be notified of the publication date in due course.

With kind regards,

Anita Estes
